# An Efficient Optimal Derivative-Free Fourth-Order Method and Its Memory Variant for Non-Linear Models and Their Dynamics

**Himani Sharma** [1], **Munish Kansal** [1,*] **and Ramandeep Behl** [2]

1   School of Mathematics, Thapar Institute of Engineering and Technology, Patiala 147004, India
2   Mathematical Modelling and Applied Computation Research Group (MMAC), Department of Mathematics, King Abdulaziz University, P.O. Box 80203, Jeddah 21589, Saudi Arabia
*   Correspondence: munish.kansal@thapar.edu

**Abstract:** We propose a new optimal iterative scheme without memory free from derivatives for solving non-linear equations. There are many iterative schemes existing in the literature which either diverge or fail to work when $f'(x) = 0$. However, our proposed scheme works even in these cases. In addition, we extended the same idea for iterative methods with memory with the help of self-accelerating parameters estimated from the current and previous approximations. As a result, the order of convergence increased from four to seven without the addition of any further functional evaluation. To confirm the theoretical results, numerical examples and comparisons with some of the existing methods are included which reveal that our scheme is more efficient than the existing schemes. Furthermore, basins of attraction are also included to describe a clear picture of the convergence of the proposed method as well as some of the existing methods.

**Keywords:** non-linear equation; iterative method with memory; *R*-order of convergence; basin of attraction

**MSC:** 65H05; 65H99

## 1. Introduction

Many problems in computational sciences and other disciplines can be modelled in the form of a non-linear equation or systems. In particular, a large number of problems in applied mathematics and engineering are solved by finding the solutions of these equations. In the literature, there are several iterative methods that have been designed by using different procedures to approximate the simple roots of a non-linear equation,

$$f(x) = 0, \tag{1}$$

where $f : I \subseteq \mathbb{R} \to \mathbb{R}$ is a real function defined in an open interval $I$. To find the roots of Equation (1), we look towards iterative schemes. A lot of iterative methods of different convergence orders already exist in the literature (see [1,2] and the references therein) to approximate the roots of Equation (1). Out of them, the most eminent one-point iterative method without memory is the quadratic convergent Newton–Raphson scheme [3] given by

$$y_n = x_n - \frac{f(x_n)}{f'(x_n)}, \; n = 0, 1, \ldots \tag{2}$$

One drawback of this method is that when $f'(x_n) = 0$, the method fails, which confines its applications. The first objective and inspiration to design iterative methods for solving this kind of problem are to obtain the highest order of convergence with the least computational cost. Therefore, a lot of researchers are interested in constructing optimal multipoint methods [4] without memory, in the sense of Kung Traub conjecture [5] which states that multipoint iterative methods without memory, requiring $n + 1$ functional

evaluations per iteration, have a convergence order at most $2^n$. Among them, an optimal fourth-order iterative method was developed by Kou et al. [6] defined by

$$
\begin{aligned}
y_n &= x_n - \frac{f(x_n)}{f'(x_n)}, \\
x_{n+1} &= x_n - \frac{f(x_n)^2 + f(y_n)^2}{f'(x_n)(f(x_n) - f(y_n))}, \quad n = 0, 1, \ldots
\end{aligned}
\tag{3}
$$

Further, Kansal et al. proposed an optimal fourth-order iterative method [7] in parameters $\alpha(\neq 1)$ and $\beta$ defined by

$$
y_n = x_n - \frac{f(x_n)}{f'(x_n)}, \quad n = 0, 1, \ldots
$$

$$
x_{n+1} = x_n - \left( \frac{\alpha + 1}{\alpha \pm \left( \dfrac{f(x_n)^2 + (\beta - 2\alpha - 2)f(x_n)f(y_n) - \beta(\alpha + 1)f(y_n)^2}{f(x_n)^2 + \beta f(x_n)f(y_n)} \right)^{1/2}} \right) \frac{f(x_n)}{f'(x_n)}.
\tag{4}
$$

Soleymani developed an optimal fourth-order method [8] given by

$$
\begin{aligned}
y_n &= x_n - \frac{f(x_n)}{f'(x_n)}, \\
x_{n+1} &= y_n - \frac{f'(x_n)^2}{f'(x_n)^2 - 2f(x_n)f(y_n)} \frac{f(y_n)}{f'(x_n)} \left( 1 + \frac{f(y_n)^2}{f(x_n)^2} \right) \left( 1 + \frac{f(y_n)^2}{f'(x_n)^2} \right) \left( 1 + \frac{f(x_n)^2}{f'(x_n)^2} \right), \\
&\quad n = 0, 1, 2, \ldots
\end{aligned}
\tag{5}
$$

Furthermore, an optimal-order method was proposed by Chun et al. [9] given by

$$
\begin{aligned}
y_n &= x_n - \frac{2}{3} \frac{f(x_n)}{f'(x_n)}, \\
x_{n+1} &= x_n + \frac{f'(x_n) + 3f'(y_n)}{2f'(x_n) - 6f'(y_n)} \frac{f(x_n)}{f'(x_n)}, \quad n = 0, 1, 2, \ldots
\end{aligned}
\tag{6}
$$

On the other hand, sometimes it is possible to increase the order of convergence without any new function evaluation based on acceleration parameter(s) which appear in the error equation of the multipoint methods without memory. It was Traub [3], who slightly altered Steffensen's method [10] and presented the first method with memory as follows:

$$
\begin{cases}
\gamma_0, x_0 \text{ are suitably given, } w_n = x_n + \gamma_n f(x_n), \ 0 \neq \gamma_n \in \mathbb{R}, \\
x_{n+1} = x_n - \frac{f(x_n)}{f[x_n, w_n]}, \quad n = 0, 1, 2, \ldots
\end{cases}
\tag{7}
$$

This method has an order of convergence of 2.414. Still, if we use a better self-accelerating parameter, there are apparent chances that the order of convergence will increase.

Following the steps of Traub, many authors are constructing higher-order methods with and without memory. Among many others, Chicharro et al. [11] presented a bi-parametric family of order four and then developed a family of methods with memory having a higher order of convergence without further increasing the number of functional evaluations per iteration. In [12], the authors presented a derivative-free form of King's family with memory. The authors in [13] developed a tri-parametric derivative-free family of Hansen–Patrick-type methods which requires only three functional evaluations to achieve optimal fourth-order convergence. Then, they extended the idea with memory as a result of which the R-order convergence increased from four to seven, without any additional functional evaluation.

The development of such methods has increased over the years. Some applications of these iterative methods can be seen in [14–17]. Thus, by taking into consideration these developments, we further attempt to propose an iterative method without memory and then convert it into a more efficient method with memory such that the order of convergence is increased without any further functional evaluation.

However, another important aspect of an iterative scheme to be considered is its stability, which is the analysis that tells us how dependent the scheme of the initial guesses used is. In this regard, a comparison between iterative methods by using the basins of attraction was developed by Ardelean [18]. This motivates us to work on the optimal-order methods and their with memory variants along with their basins of attraction.

The rest of the paper is organized as follows. Section 2 contains the development of a new iterative method without memory and the proof of its order of convergence. Section 3 covers the inclusion of memory to develop a new iterative method with memory and its error analysis. Numerical results for the proposed methods and comparisons with some of the existing methods to illustrate our theoretical results are given in Section 4. Section 5 depicts the convergence of the methods using basins of attraction. Lastly, Section 6 collates the conclusions.

*R-Order of Convergence*

For finding the R-order convergence [19] of our proposed method with memory, we make use of the following Theorem 1 given by Traub.

**Theorem 1.** *Suppose that* $(IM)$ *is an iterative method with memory that generates a sequence* $\{x_m\}$ *(converging to the root* $\xi$*) of approximations to* $\xi$*. If there exists a non-zero constant* $\zeta$ *and non-negative numbers* $s_j$*,* $0 \leq j \leq k$*, such that the inequality,*

$$| \epsilon_{m+1} | \leq \zeta \prod_{j=0}^{k} | \epsilon_{m-j} |^{s_j}$$

*holds, then the R-order of convergence of the iterative method* $(IM)$ *satisfies the inequality,*

$$O_R((IM), \xi) \geq t^*,$$

*where* $t^*$ *is the unique positive root of the equation,*

$$t^{k+1} - \sum_{j=0}^{k} s_j t^{k-j} = 0. \tag{8}$$

## 2. Iterative Method without Memory and Its Convergence Analysis

We aim to construct a new two-point derivative-free optimal scheme without memory in this section and extend it to a memory scheme.

If the well-known Steffensen's method is combined with Newton's method, we obtain the following fourth-order scheme:

$$\begin{cases} y_n &= x_n - \dfrac{f(x_n)^2}{f(w_n) - f(x_n)}, \\ x_{n+1} &= y_n - \dfrac{f(y_n)}{f'(y_n)}, \end{cases} \tag{9}$$

where $w_n = x_n + f(x_n)$. To avoid the computation of $f'(y_n)$, the authors in [20] approximated it by the derivative $m'(y_n)$ of the following first-degree Padé approximant:

$$m(t) = \frac{a_1 + a_2(t - y_n)}{1 + a_3(t - y_n)}, \tag{10}$$

where $a_1$, $a_2$ and $a_3$ are real parameters to be determined satisfying the following conditions:

$$m(x_n) = f(x_n), \tag{11}$$
$$m(y_n) = f(y_n), \tag{12}$$
$$m(w_n) = f(w_n). \tag{13}$$

Using these conditions, the derivative of the Padé approximant evaluated in $y_n$ is given as

$$m'(y_n) = \frac{f[x_n, y_n]f[y_n, w_n]}{f[x_n, w_n]}. \tag{14}$$

Using (14) in the second step of (9), they presented the following scheme:

$$\begin{cases} y_n &= x_n - \dfrac{f(x_n)^2}{f(w_n) - f(x_n)}, \\ x_{n+1} &= y_n - \dfrac{f(y_n)f[x_n, w_n]}{f[x_n, y_n]f[y_n, w_n]}, \end{cases} \tag{15}$$

where $w_n = x_n + f(x_n)$. This scheme is optimal in the sense of the Kung–Traub conjecture having an order of convergence of four with three functional evaluations per iteration, $f(x_n)$, $f(y_n)$ and $f(w_n)$.

Now, in order to extend the method with memory, we devise the idea of introducing two parameters $\gamma$ and $\lambda$ in (15) and we present a modification in this method as follows:

$$\begin{cases} y_n &= x_n - \dfrac{f(x_n)}{f[x_n, w_n] + \lambda f(w_n)}, \\ x_{n+1} &= y_n - \dfrac{f(y_n)(f[x_n, w_n] + \lambda f(w_n))}{(f[x_n, y_n] + \lambda f(w_n))f[y_n, w_n]}, \end{cases} \tag{16}$$

where $w_n = x_n + \gamma f(x_n)$.

This modified scheme yields the optimal order of convergence 4 having three functional evaluations per iteration, $f(x_n)$, $f(y_n)$ and $f(w_n)$.

Next, we establish the convergence results for our proposed family without memory given by Equation (16).

**Theorem 2.** *Suppose that $f : D \subset \mathbb{R} \to \mathbb{R}$ is a real function suitably differentiable in a domain $D$. If $\xi \in D$ is a simple root of $f(x) = 0$ and an initial guess $x_0$ is sufficiently close to $\xi$, then the iterative method given by Equation (16), converges to $\xi$ with convergence order $p = 4$ having the following error relation,*

$$e_{n+1} = (1 + f'(\xi)\gamma)^2(\lambda + c_2)\left((2 + f'(\xi)\gamma)\lambda c_2 + 2c_2^2 - c_3\right)e_n^4 + O(e_n)^5,$$

*where $e_n = x_n - \xi$, $\xi$ is a simple root of $f(x) = 0$ and $c_n = \dfrac{f^{(n)}(\xi)}{n!f'(\xi)}$, $n = 2, 3, \ldots$*

**Proof.** Expanding $f(x_n)$ about $x_n = \xi$ by the Taylor series, we have

$$f(x_n) = f'(\xi)(e_n + c_2 e_n^2 + c_3 e_n^3 + c_4 e_n^4) + O(e_n)^5. \tag{17}$$

Using Equation (17) in the first step of Equation (16), we have

$$
\begin{aligned}
e_{n,y} = y_n - \xi =\ & (1 + f'(\xi)\gamma)(\lambda + c_2)e_n^2 + (-(2 + 2f'(\xi)\gamma + f'(\xi)^2\gamma^2)\lambda c_2 - (2 + \\
& 2f'(\xi)\gamma + f'(\xi)^2\gamma^2)c_2^2 - (1 + f'(\xi)\gamma)((1 + f'(\xi)\gamma)\lambda^2 - (2 + \\
& f'(\xi)\gamma)c_3))e_n^3 + ((5 + 7f'(\xi)\gamma + 4f'(\xi)^2\gamma^2 + f'(\xi)^3\gamma^3)\lambda c_2^2 + \\
& (4 + 5f'(\xi)\gamma + 3f'(\xi)^2\gamma^2 + f'(\xi)^3\gamma^3)c_2^3 - (4 + 7f'(\xi)\gamma + \\
& 5f'(\xi)^2\gamma^2 + f'(\xi)^3\gamma^3)\lambda c_3 - c_2(-(3 + 5f'(\xi)\gamma + 3f'(\xi)^2\gamma^2 + \\
& f'(\xi)^3\gamma^3)\lambda^2 + (7 + 10f'(\xi)\gamma + 7f'(\xi)^2\gamma^2 + 2f'(\xi)^3\gamma^3)c_3) + \\
& (1 + f'(\xi)\gamma)((1 + f'(\xi)\gamma)^2\lambda^3 + (3 + 3f'(\xi)\gamma + f'(\xi)^2\gamma^2)c_4))e_n^4 + \\
& O(e_n)^5.
\end{aligned}
\tag{18}
$$

In addition, the Taylor's expansion of $f(y_n)$ is

$$
f(y_n) = f'(\xi)(e_{n,y} + c_2 e_{n,y}^2 + c_3 e_{n,y}^3 + c_4 e_{n,y}^4) + O(e_{n,y})^5.
\tag{19}
$$

Using Equations (17)–(19), we have

$$
\begin{aligned}
\frac{f(y_n)(f[x_n, w_n] + \lambda f(w_n))}{(f[x_n, y_n] + \lambda f(w_n))f[y_n, w_n]} =\ & (1 + f'(\xi)\gamma)(\lambda + c_2)e_n^2 + (-(2 + 2f'(\xi)\gamma + \\
& f'(\xi)^2\gamma^2)\lambda c_2 - (2 + 2f'(\xi)\gamma + f'(\xi)^2\gamma^2)c_2^2 - (1 + \\
& f'(\xi)\gamma)((1 + f'(\xi)\gamma)\lambda^2 - (2 + f'(\xi)\gamma)c_3))e_n^3 + ((1 - \\
& 2f'(\xi)\gamma - 2f'(\xi)^2\gamma^2)\lambda c_2^2 + (2 + f'(\xi)\gamma + f'(\xi)^2\gamma^2 + \\
& f'(\xi)^3\gamma^3)c_2^3 - (3 + 5f'(\xi)\gamma + 4f'(\xi)^2\gamma^2 + \\
& f'(\xi)^3\gamma^3)\lambda c_3 - c_2((-1 + f'(\xi)^2\gamma^2)\lambda^2 + 2(3 + \\
& 4f'(\xi)\gamma + 3f'(\xi)^2\gamma^2 + f'(\xi)^3\gamma^3)c_3) + (1 + \\
& f'(\xi)\gamma)((1 + f'(\xi)\gamma)^2\lambda^3 + (3 + 3f'(\xi)\gamma + \\
& f'(\xi)^2\gamma^2)c_4))e_n^4 + O(e_n)^5.
\end{aligned}
\tag{20}
$$

Finally, putting Equation (20) into the second step of Equation (16), we obtain

$$
e_{n+1} = (1 + f'(\xi)\gamma)^2(\lambda + c_2)\left((2 + f'(\xi)\gamma)\lambda c_2 + 2c_2^2 - c_3\right)e_n^4 + O(e_n)^5,
\tag{21}
$$

which is the error equation for the proposed optimal scheme given by Equation (16) with a convergence order of four. This completes the proof. $\quad\square$

## 3. Iterative Method with Memory and Its Convergence Analysis

Now, we present an extension to the method given by Equation (16) by the inclusion of memory to improve the convergence order without the addition of any new functional evaluations.

If we clearly observed, it can be seen from the error relation given in Equation (21) that the order of convergence of the proposed family given by Equation (16) is 4 if $\gamma \neq \dfrac{-1}{f'(\xi)}$ and $\lambda \neq -c_2$. Therefore, if $\gamma = \dfrac{-1}{f'(\xi)}$ and $\lambda = -c_2 = -\dfrac{f''(\xi)}{2f'(\xi)}$, then the order of convergence of our proposed family can be improved, but this value cannot be reached because the values of $f'(\xi)$ and $f''(\xi)$ are not practically available. Instead, we can use approximations calculated by already available information [21]. Hence, the main idea in constructing the methods with memory consists of the calculation of parameters $\gamma = \gamma_n$ and $\lambda = \lambda_n$ as the iteration proceeds by the formulae,

$$
\gamma_n = \frac{-1}{f'(\xi)} \quad \text{and} \quad \lambda_n = -c_2 = -\frac{f''(\xi)}{2f'(\xi)}
$$

for $n = 1, 2, \ldots$ Further, it is also assumed that the initial estimates $\gamma_0$ and $\lambda_0$ must be chosen before starting the iterations. Thus, we give an estimation for $\gamma_n$ and $\lambda_n$ given by

$$\gamma_n = \frac{-1}{N_3'(x_n)} \quad \text{and} \quad \lambda_n = \frac{-N_4''(w_n)}{2N_4'(w_n)}, \tag{22}$$

where $N_3(k) = N_3(k; x_n, x_{n-1}, y_{n-1}, w_{n-1})$ and $N_4(k) = N_4(k; w_n, x_n, w_{n-1}, y_{n-1}, x_{n-1})$ are Newton's interpolating polynomials of the third- and fourth-degrees, respectively, which are set through the best available nodal points, $(x_n, x_{n-1}, y_{n-1}, w_{n-1})$ for $N_3$ and $(w_n, x_n, w_{n-1}, y_{n-1}, x_{n-1})$ for $N_4$.

Thus, by replacing $\gamma$ by $\gamma_n$ and $\lambda$ by $\lambda_n$ in the method given by Equation (16), we obtain a new family with memory as follows:

$$\begin{cases} \gamma_0, \ \lambda_0, \ x_0 \text{ are given, } w_0 = x_0 + \gamma_0 f(x_0) \\ \gamma_n = \frac{-1}{N_3'(x_n)}, \ w_n = x_n + \gamma_n f(x_n), \ \lambda_n = \frac{-N_4''(w_n)}{2N_4'(w_n)}, \ n = 1, 2, \ldots, \\ y_n = x_n - \frac{f(x_n)}{f[x_n, w_n] + \lambda_n f(w_n)}, \\ x_{n+1} = y_n - \frac{f(y_n)(f[x_n, w_n] + \lambda_n f(w_n))}{(f[x_n, y_n] + \lambda_n f(w_n))f[y_n, w_n]}. \end{cases} \tag{23}$$

Next, we establish the convergence results for our proposed family with memory given by Equation (23).

**Theorem 3.** *Suppose that $f : D \subset \mathbb{R} \to \mathbb{R}$ is a real function suitably differentiable in a domain $D$. If $\xi \in D$ is a simple root of $f(x) = 0$ and an initial guess $x_0$ is sufficiently close to $\xi$, then the iterative method given by Equation (23) converges to $\xi$ with a convergence order of at least 7.*

**Proof.** Let $\{x_n\}$ be a sequence of approximations generated by an iterative method ($IM$). If this sequence converges to zero $\xi$ of $f$ with the $R$-order $(\geq r)$ of $IM$, then we can write

$$e_{n+1} \sim D_{n,r} e_n^r, \ e_n = x_n - \xi, \tag{24}$$

where $D_{n,r}$ tends to the asymptotic error constant $D_r$ of $IM$, when $n \to \infty$. Thus,

$$e_{n+1} \sim D_{n,r}(D_{n-1,r}e_{n-1}^r)^r = D_{n,r}D_{n-1,r}^r e_{n-1}^{r^2} \tag{25}$$

Let the iterative sequences $\{w_n\}$ and $\{yn\}$ have $R$-orders $r_1$ and $r_2$, respectively. Therefore, we obtain

$$e_{n,w} = w_n - \xi \sim D_{n,r_1} e_n^{r_1} \sim D_{n,r_1}(D_{n-1,r}e_{n-1}^r)^{r_1} = D_{n,r_1}D_{n-1,r}^{r_1} e_{n-1}^{rr_1} \tag{26}$$

and

$$e_{n,y} = y_n - \xi \sim D_{n,r_2} e_n^{r_2} \sim D_{n,r_2}(D_{n-1,r}e_{n-1}^r)^{r_2} = D_{n,r_2}D_{n-1,r}^{r_2} e_{n-1}^{rr_2}. \tag{27}$$

Using (26), (27) and a lemma stated in [13], we obtain

$$\begin{aligned} 1 + \gamma_n f'(\xi) &\sim \psi_1 e_{n-1,w} e_{n-1,y} e_{n-1} = \psi_1 D_{n-1,r_1} D_{n-1,r_2} e_{n-1}^{r_1+r_2+1}, \\ \lambda_n + c_2 &\sim \psi_2 e_{n-1,w} e_{n-1,y} e_{n-1} = \psi_2 D_{n-1,r_1} D_{n-1,r_2} e_{n-1}^{r_1+r_2+1}. \end{aligned} \tag{28}$$

In view of our proposed family of methods without memory given by Equation (16), we have the following error relations,

$$e_{n,w} = (1 + \gamma f'(\xi))e_n + O(e_n)^2, \tag{29}$$

$$e_{n,y} = (1 + \gamma f'(\xi))(\lambda + c_2)e_n^2 + O(e_n)^3, \tag{30}$$

$$e_{n+1} = \phi_1(1 + \gamma f'(\xi))^2(\lambda + c_2)e_n^4 + O(e_n)^5, \tag{31}$$

where $\phi_1 = (2 + f'(\xi)\gamma)\lambda c_2 + 2c_2^2 - c_3$.

According to the error relations given by Equations (29)–(31) with self-accelerating parameters, $\gamma = \gamma_n$ and $\lambda = \lambda_n$, we can write the corresponding error relations for the methods given by Equation (23) with memory as follows:

$$e_{n,w} \sim (1 + \gamma_n f'(\xi))e_n, \tag{32}$$

$$e_{n,y} \sim (1 + \gamma_n f'(\xi))(\lambda_n + c_2)e_n^2, \tag{33}$$

$$e_{n+1} \sim \phi_2(1 + \gamma_n f'(\xi))^2(\lambda_n + c_2)e_n^4, \tag{34}$$

where $\phi_2 = (2 + f'(\xi)\gamma_n)\lambda_n c_2 + 2c_2^2 - c_3$ depending on iteration index since $\gamma_n$ and $\lambda_n$ are re-calculated in each step. Now using Equations (28) and (32)–(34), we obtain the following relations:

$$e_{n,w} \sim (1 + \gamma_n f'(\xi))e_n \sim \psi_1 D_{n-1,r_1} D_{n-1,r_2} D_{n-1,r} e_{n-1}^{r+r_1+r_2+1}, \tag{35}$$

$$e_{n,y} \sim (1 + \gamma_n f'(\xi))(\lambda_n + c_2)e_n^2 \sim \psi_1 \psi_2 D_{n-1,r_1}^2 D_{n-1,r_2}^2 D_{n-1,r}^2 e_{n-1}^{2r+2r_1+2r_2+2}, \tag{36}$$

$$e_{n+1} \sim \phi_2(1 + \gamma_n f'(\xi))^2(\lambda_n + c_2)e_n^4 \sim \phi_2 \psi_1^2 \psi_2 D_{n-1,r_1}^3 D_{n-1,r_2}^3 D_{n-1,r}^4 e_{n-1}^{4r+3r_1+3r_2+3}. \tag{37}$$

Now, comparing the error exponents of $e_{n-1}$ on the right-hand side of the pairs given by Equations (26) with (35), (27) with (36) and (25) with (37), respectively, we obtain the following system of equations:

$$
\begin{aligned}
rr_1 - r - r_1 - r_2 &= 1, \\
rr_2 - 2r - 2r_1 - 2r_2 &= 2, \\
r^2 - 4r - 3r_1 - 3r_2 &= 3.
\end{aligned}
\tag{38}
$$

Solving this system of equations, we obtain a non-trivial solution as $r_1 = 2$, $r_2 = 4$ and $r = 7$. Hence, we can conclude that the lower bound of the R-order of our proposed family with memory given by Equation (23) is seven. This completes our proof. □

## 4. Numerical Results

In this section, the numerical results of our proposed scheme are examined. Furthermore, we will demonstrate the corresponding results after comparison with some existing schemes, both with and without memory. All calculations have been accomplished using Mathematica 11.1 in multiple precision arithmetic environments with specification of a processor Intel(R) Core(TM) i5-1035G1 CPU @ 1.00 GHz 1.20 GHz (64-bit operating system), Windows 11. We suppose that the initial values of $\gamma$ (or $\gamma_0$) and $\lambda$ (or $\lambda_0$) must be selected prior to performing the iterations and a suitable $x_0$ be given.

The functions used for our computations are given in Table 1.

**Table 1.** Test functions along with their roots and initial guesses taken.

| Test Function | Real Root | Initial Guess Taken |
|---|---|---|
| $f_1(x) = (x - 2)(x^{10} + x + 1)e^{-x-1} = 0$ | 2 | 1.925 |
| $f_2(x) = e^{x^2+7x-30} - 1 = 0$ | 3 | 2.90 |
| $f_3(x) = \sin(\pi x)e^{x^2+x\cos x-1} + x\log(x\sin x + 1) = 0$ | 0 | 0.05 |
| $f_4(x) = e^{x^3-x} - \cos(x^2 - 1) + x^3 + 1 = 0$ | $-1$ | $-1.10$ |
| $f_5(x) = e^{x^2-3x}\sin x + \log(x^2 + 1) = 0$ | 0 | 0.05 |

To check the theoretical order of convergence, the computational order of convergence [22], $\rho_c$ (COC) is calculated using the following formula,

$$\rho_c = \frac{log(|f(x_k)/f(x_{k-1})|)}{log(|f(x_{k-1})/f(x_{k-2})|)}, \quad k = 2, 3, \ldots,$$

considering the last three approximations in the iterative procedure. The errors of approximations to the respective zeros of the test functions, $\mid x_n - \xi \mid$ and COC are displayed in Tables 2 and 3.

**Table 2.** Comparison of the different methods without memory.

| Without Memory Methods | $\mid x_1 - \xi \mid$ | $\mid x_2 - \xi \mid$ | $\mid x_3 - \xi \mid$ | $\rho_c$ | CPU Time |
|---|---|---|---|---|---|
| $f_1(x)$ | | | | | |
| $PM(\gamma = -0.1, \lambda = 0.1)$ | $1.1026 \times 10^{-2}$ | $3.4683 \times 10^{-5}$ | $2.3844 \times 10^{-15}$ | 4.0308 | 0.390 |
| $SM(\alpha = 10, \gamma = -0.01)$ | $4.5722 \times 10^{-2}$ | $1.4814 \times 10^{-3}$ | $1.8466 \times 10^{-10}$ | 4.8888 | 0.343 |
| $AM_1$ | F | F | F | ## | – |
| $CM$ | $6.6406 \times 10^{-2}$ | $1.8454 \times 10^{-3}$ | $3.2406 \times 10^{-9}$ | 3.4574 | 0.329 |
| $f_2(x)$ | | | | | |
| $PM(\gamma = -0.1, \lambda = 0.1)$ | $5.3295 \times 10^{-3}$ | $3.6701 \times 10^{-8}$ | $6.3025 \times 10^{-29}$ | 4.0108 | 0.265 |
| $SM(\alpha = 10, \gamma = -0.01)$ | F | F | F | ## | – |
| $AM_1$ | F | F | F | ## | – |
| $CM$ | NC | NC | NC | # | – |
| $f_3(x)$ | | | | | |
| $PM(\gamma = -0.1, \lambda = 0.1)$ | $7.6728 \times 10^{-6}$ | $4.3783 \times 10^{-21}$ | $4.6420 \times 10^{-82}$ | 4.0000 | 0.671 |
| $SM(\alpha = 10, \gamma = -0.01)$ | $2.2439 \times 10^{-5}$ | $1.4028 \times 10^{-18}$ | $2.1427 \times 10^{-71}$ | 4.0000 | 0.875 |
| $AM_1$ | $3.8672 \times 10^{-5}$ | $1.3302 \times 10^{-17}$ | $1.8622 \times 10^{-67}$ | 4.0000 | 0.812 |
| $CM$ | $2.2767 \times 10^{-5}$ | $1.1497 \times 10^{-18}$ | $7.4781 \times 10^{-72}$ | 4.0000 | 0.624 |
| $f_4(x)$ | | | | | |
| $PM(\gamma = -0.1, \lambda = 0.1)$ | $3.6861 \times 10^{-6}$ | $1.6522 \times 10^{-23}$ | $6.6701 \times 10^{-93}$ | 4.0000 | 0.312 |
| $SM(\alpha = 10, \gamma = -0.01)$ | $1.4106 \times 10^{-5}$ | $2.4856 \times 10^{-21}$ | $2.3942 \times 10^{-84}$ | 4.0000 | 0.453 |
| $AM_1$ | $9.0450 \times 10^{-5}$ | $1.2109 \times 10^{-15}$ | $3.8809 \times 10^{-59}$ | 4.0001 | 0.422 |
| $CM$ | $2.2615 \times 10^{-5}$ | $1.8131 \times 10^{-19}$ | $7.4932 \times 10^{-76}$ | 4.0000 | 0.281 |
| $f_5(x)$ | | | | | |
| $PM(\gamma = -0.1, \lambda = 0.1)$ | $1.0074 \times 10^{-5}$ | $3.6243 \times 10^{-20}$ | $6.0724 \times 10^{-78}$ | 4.0000 | 0.390 |
| $SM(\alpha = 10, \gamma = -0.01)$ | $3.8032 \times 10^{-4}$ | $1.0334 \times 10^{-12}$ | $5.6176 \times 10^{-47}$ | 4.0003 | 0.594 |
| $CM$ | $1.6301 \times 10^{-4}$ | $2.0715 \times 10^{-14}$ | $5.4018 \times 10^{-54}$ | 3.9999 | 0.359 |

*F*—Method fails; *##*—COC not required in case of failure; *NC*—Not converging to root after three iterations; *#*—COC not mentioned in case of non-convergence after three iterations.

**Table 3.** Comparison of the different methods with memory.

| Without Memory Methods | $\mid x_1 - \xi \mid$ | $\mid x_2 - \xi \mid$ | $\mid x_3 - \xi \mid$ | $\rho_c$ | CPU Time |
|---|---|---|---|---|---|
| $f_1(x)$ | | | | | |
| $PMM(\gamma_0 = -0.1, \lambda_0 = 0.1)$ | $1.1025 \times 10^{-2}$ | $2.0090 \times 10^{-11}$ | $1.7059 \times 10^{-72}$ | 6.9728 | 0.984 |
| $AM_2(\gamma_0 = \lambda_0 = 0.1)$ | $3.7765 \times 10^{-1}$ | $1.8449 \times 10^{-2}$ | $6.9515 \times 10^{-12}$ | 4.8678 | 1.031 |
| $DM_1(\gamma_0 = \lambda_0 = 0.1)$ | NC | NC | NC | # | – |
| $DM_2(\gamma_0 = \lambda_0 = 0.1)$ | $9.4868 \times 10^{-1}$ | $7.6918 \times 10^{-2}$ | $3.7808 \times 10^{-6}$ | 1.9871 | 0.969 |
| $f_2(x)$ | | | | | |
| $PMM(\gamma_0 = -0.1, \lambda_0 = 0.1)$ | $5.3295 \times 10^{-3}$ | $2.2157 \times 10^{-12}$ | $6.5108 \times 10^{-78}$ | 6.9741 | 0.844 |
| $AM_2(\gamma_0 = \lambda_0 = 0.1)$ | $5.1899 \times 10^{-2}$ | $3.2288 \times 10^{-6}$ | $4.5631 \times 10^{-35}$ | 6.6121 | 0.812 |
| $DM_1(\gamma_0 = \lambda_0 = 0.1)$ | F | F | F | ## | – |
| $DM_2(\gamma_0 = \lambda_0 = 0.1)$ | NC | NC | NC | # | – |

**Table 3.** *Cont.*

| Without Memory Methods | $\|x_1 - \xi\|$ | $\|x_2 - \xi\|$ | $\|x_3 - \xi\|$ | $\rho_c$ | CPU Time |
|---|---|---|---|---|---|
| $f_3(x)$ | | | | | |
| $PMM(\gamma_0 = -0.1, \lambda_0 = 0.1)$ | $7.6728 \times 10^{-6}$ | $4.8557 \times 10^{-38}$ | $7.5120 \times 10^{-261}$ | 6.9199 | 3.047 |
| $AM_2(\gamma_0 = \lambda_0 = 0.1)$ | $4.2993 \times 10^{-6}$ | $1.1962 \times 10^{-37}$ | $2.2842 \times 10^{-258}$ | 6.9946 | 3.047 |
| $DM_1(\gamma_0 = \lambda_0 = 0.1)$ | $2.1772 \times 10^{-5}$ | $7.2683 \times 10^{-34}$ | $6.9858 \times 10^{-232}$ | 6.9537 | 3.141 |
| $DM_2(\gamma_0 = \lambda_0 = 0.1)$ | $1.2537 \times 10^{-5}$ | $3.6538 \times 10^{-36}$ | $5.6673 \times 10^{-248}$ | 6.9365 | 3.266 |
| $f_4(x)$ | | | | | |
| $PMM(\gamma_0 = -0.1, \lambda_0 = 0.1)$ | $3.6861 \times 10^{-6}$ | $2.9711 \times 10^{-39}$ | $2.0613 \times 10^{-271}$ | 7.0152 | 1.360 |
| $AM_2(\gamma_0 = \lambda_0 = 0.1)$ | $1.2532 \times 10^{-5}$ | $2.6367 \times 10^{-35}$ | $6.0850 \times 10^{-244}$ | 7.0303 | 1.328 |
| $DM_1(\gamma_0 = \lambda_0 = 0.1)$ | $1.2723 \times 10^{-5}$ | $2.8862 \times 10^{-35}$ | $1.1458 \times 10^{-243}$ | 7.0301 | 1.359 |
| $DM_2(\gamma_0 = \lambda_0 = 0.1)$ | $1.2656 \times 10^{-5}$ | $2.7964 \times 10^{-35}$ | $9.1836 \times 10^{-244}$ | 7.0301 | 1.358 |
| $f_5(x)$ | | | | | |
| $PMM(\gamma_0 = -0.1, \lambda_0 = 0.1)$ | $1.0074 \times 10^{-5}$ | $6.5505 \times 10^{-34}$ | $9.9064 \times 10^{-231}$ | 6.9827 | 1.625 |
| $AM_2(\gamma_0 = \lambda_0 = 0.1)$ | $2.5921 \times 10^{-5}$ | $6.2077 \times 10^{-31}$ | $4.4988 \times 10^{-211}$ | 7.0310 | 1.672 |
| $DM_1(\gamma_0 = \lambda_0 = 0.1)$ | $6.0285 \times 10^{-5}$ | $6.9376 \times 10^{-28}$ | $9.7865 \times 10^{-190}$ | 7.0557 | 1.891 |
| $DM_2(\gamma_0 = \lambda_0 = 0.1)$ | $1.2734 \times 10^{-5}$ | $3.1600 \times 10^{-32}$ | $3.9836 \times 10^{-220}$ | 7.0625 | 1.812 |

*F*—Method fails; ##—COC not required in case of failure; *NC*—Not converging to root after three iterations; #—COC not mentioned in case of non-convergence after three iterations.

We consider the following existing methods for the comparisons:
Soleymani et al. method (*SM*) without memory [23]:

$$y_n = x_n - \frac{f(x_n)}{f[x_n, w_n]}, \quad w_n = x_n + \gamma f(x_n), \quad \gamma \in \mathbb{R} \backslash \{0\},$$

$$x_{n+1} = x_n - \left(\frac{f(x_n) + f(y_n)}{f[x_n, w_n]}\right) - \left(\frac{2f(x_n) + \alpha f(y_n)}{f[x_n, w_n]}\right)\left(\frac{f(y_n)}{f(x_n)}\right)^2\left(1 - \frac{\gamma f[x_n, w_n]}{2 + 2\gamma f[x_n, w_n]}\right), \quad \alpha \in \mathbb{R}, \quad (39)$$

$$n = 0, 1, 2, \ldots$$

Cordero et al. method ($AM_1$) without memory [20]:

$$y_n = x_n - \frac{f(x_n)}{f[x_n, w_n]}, \quad w_n = x_n + f(x_n),$$

$$x_{n+1} = y_n - \frac{f(y_n)f[x_n, w_n]}{f[x_n, y_n]f[y_n, w_n]}, \quad n = 0, 1, 2, \ldots \quad (40)$$

Chun method (*CM*) without memory [24]:

$$y_n = x_n - \frac{f(x_n)}{f'(x_n)},$$

$$x_{n+1} = x_n - \frac{f(x_n)}{f'(x_n)}(1 + u + 2u^2), \quad u = \frac{f(y_n)}{f(x_n)} \quad n = 0, 1, 2, \ldots \quad (41)$$

Cordero et al. method ($AM_2$) with memory [25]:

$$\gamma_0, \ \lambda_0, \ x_0 \text{ are given}, \ w_0 = x_0 + \gamma_0 f(x_0)$$

$$\gamma_n = \frac{-1}{N_3'(x_n)}, \quad w_n = x_n + \gamma_n f(x_n), \quad \lambda_n = \frac{-N_4''(w_n)}{2N_4'(w_n)}, \quad n = 1, 2, \ldots,$$

$$y_n = x_n - \frac{f(x_n)}{f[x_n, w_n] + \lambda_n f(w_n)}, \quad (42)$$

$$x_{n+1} = y_n - \frac{f(y_n)}{(f[x_n, y_n] + (y_n - x_n)f[x_n, w_n, y_n]},$$

where $N_3$ and $N_4$ are as defined in Section 3.

Džunić method ($DM_1$ and $DM_2$) with memory [26]:

$$\gamma_0, \lambda_0, x_0 \text{ are given, } w_0 = x_0 + \gamma_0 f(x_0)$$

$$\gamma_n = \frac{-1}{N_3'(x_n)}, \quad w_n = x_n + \gamma_n f(x_n), \quad \lambda_n = \frac{-N_4''(w_n)}{2N_4'(w_n)}, \quad n = 1, 2, \ldots,$$

$$y_n = x_n - \frac{f(x_n)}{f[x_n, w_n] + \lambda_n f(w_n)},$$

$$x_{n+1} = y_n - \frac{f(y_n)g(t_n)}{(f[y_n, w_n] + \lambda_n f(w_n)}, \quad t_n = \frac{f(y_n)}{f(x_n)},$$

(43)

where $N_3$ and $N_4$ are as defined in Section 3.

Furthermore, we consider some real-life problems, which are as follows:

**Example 1.** *Fractional conversion in a chemical reactor [27],*

$$f_6(x) = \frac{x}{1-x} - 5 \log \frac{0.4(1-x)}{0.4 - 0.5x} + 4.45977 = 0. \tag{44}$$

*Here, $x$ denotes the fractional conversion of quantities in a chemical reactor. If $x$ is less than zero or greater than one, then the above fractional conversion will be of no physical meaning. Hence, $x$ is taken to be bounded in the region $0 \le x \le 1$. Moreover, the desired root is $\xi \approx 0.7573962462537538$.*

**Example 2.** *The path traversed by an electron in the air gap between two parallel plates considering the multi-factor effect is given by*

$$u(t) = u_0 + \left( v_0 + c_0 \frac{E}{m\omega} \sin \omega t_0 + \beta \right)(t - t_0) + c_0 \frac{E_0}{m\omega^2} (\cos(\omega t + \beta) + \sin(\omega t + \beta)), \tag{45}$$

*where $u_0$ and $v_0$ are the position and velocity of the electron at time $t_0$, $m$ and $c_0$ are the mass and charge of the electron at rest and $E_0 \sin(\omega t + \beta)$ is the RF electric field between the plates. If particular parameters are chosen, Equation (45) can be simplified as*

$$f_7(x) = x - \frac{1}{2} \cos x + \frac{\pi}{4} = 0. \tag{46}$$

*The desired root of Equation (46) is $\xi \approx -0.3090932715417949$.*

We also implemented our proposed schemes given by Equations (16) and (23) on the above-mentioned problems. Tables 4 and 5 demonstrate the corresponding results. Further, Table 2 demonstrates COC for our proposed method without memory ($PM$) given by Equation (16), the method given by Equation (39) denoted as $SM$, the method given by Equation (40) denoted as $AM_1$, and the method given by Equation (41) denoted as $CM$, respectively. Table 3 demonstrates COC for our proposed method with memory ($PMM$) given by Equation (23), the method given by Equation (42) denoted as $AM_2$, and the method given by Equation (43) by taking $g(t) = 1 + t$ denoted as $DM_1$ and $g(t) = 1/(1-t)$ denoted by $DM_2$, respectively.

It can be seen from Tables 2 and 3 that for the function $f_1$, $AM_1$ fails to provide a solution and $DM_1$ requires more than three iterations to converge to the root. Furthermore, $PMM$ converges to the desired root with an error of approximations much lower than $AM_2$ and $DM_2$. For the function $f_2$, $SM$, $AM_1$ and $DM_1$ fail to provide a solution and $CM$ and $DM_2$ do not converge to the desired solution within three iterations. $SM$ has a somewhat complex structure, and as a consequence takes more time than our method $PM$ in most of the cases to converge to the root. Furthermore, $AM$ and $DM_2$ converge to the root taking more time than $PM$ and $PMM$, respectively. $CM$ has a drawback of its derivative, so it will not work on points at which the function is zero or close to zero.

Furthermore, for functions $f_3$, $f_4$ and $f_5$, the proposed methods *PM* and *PMM* converge to the required root with minimum error compared to the existing methods.

Hence, we can conclude that our methods work on several functions to obtain roots, whereas the existing methods have some limitations.

**Remark 1.** *The proposed schemes given by Equations (16) and (23) have been compared to some already existing methods and it can be seen from the computational results that our proposed schemes give results in many cases where the existing methods fail in terms of COC and errors, as depicted in Tables 2–5. Our methods display a noticeable decrease in approximation errors, as shown in the above-mentioned tables.*

**Remark 2.** *From Tables 4 and 5, one can observe that for the function $f_6$, the existing method $AM_1$ fails to converge. In addition, for the function $f_7$, an obvious decrease in the order of convergence of the existing methods is noticeable.*

**Table 4.** Comparison of the different methods without memory for real-life problems.

| Without Memory Methods | $\mid x_1 - \xi \mid$ | $\mid x_2 - \xi \mid$ | $\mid x_3 - \xi \mid$ | $\rho_c$ | CPU Time |
|---|---|---|---|---|---|
| $f_6(x)$ | | | | | |
| $PM(\gamma = -0.1, \lambda = 0.1)$ | $7.5452 \times 10^{-3}$ | $1.0390 \times 10^{-3}$ | $3.8220 \times 10^{-7}$ | 3.7581 | 0.454 |
| $SM(\alpha = 10, \gamma = -0.01)$ | $1.4049 \times 10^{-3}$ | $5.3743 \times 10^{-7}$ | $8.8482 \times 10^{-17}$ | 4.0239 | 0.390 |
| $AM_1$ | F | F | F | ## | – |
| $CM$ | $1.0275 \times 10^{-3}$ | $1.7055 \times 10^{-8}$ | $8.8493 \times 10^{-17}$ | 3.9915 | 0.265 |
| $f_7(x)$ | | | | | |
| $PM(\gamma = -0.1, \lambda = 0.1)$ | $1.0994 \times 10^{-3}$ | $8.4592 \times 10^{-14}$ | $3.0463 \times 10^{-31}$ | 3.9999 | 0.281 |
| $SM(\alpha = 10, \gamma = -0.01)$ | $8.6465 \times 10^{-4}$ | $6.5137 \times 10^{-14}$ | $3.0463 \times 10^{-31}$ | 4.0001 | 0.374 |
| $AM_1$ | $2.3818 \times 10^{-3}$ | $3.9429 \times 10^{-12}$ | $3.0463 \times 10^{-31}$ | 3.9998 | 0.405 |
| $CM$ | $1.5968 \times 10^{-3}$ | $6.6431 \times 10^{-13}$ | $3.0463 \times 10^{-31}$ | 3.9998 | 0.219 |

*F*—Method fails; ##—COC not required in case of failure.

**Table 5.** Comparison of the different methods with memory for real-life problems.

| Without Memory Methods | $\mid x_1 - \xi \mid$ | $\mid x_2 - \xi \mid$ | $\mid x_3 - \xi \mid$ | $\rho_c$ | CPU Time |
|---|---|---|---|---|---|
| $f_6(x)$ | | | | | |
| $PMM(\gamma_0 = -0.1, \lambda_0 = 0.1)$ | $7.4286 \times 10^{-3}$ | $9.0440 \times 10^{-8}$ | $8.8493 \times 10^{-17}$ | 7.1919 | 1.641 |
| $AM_2(\gamma_0 = \lambda_0 = 0.1)$ | $3.4817 \times 10^{-4}$ | $1.7393 \times 10^{-13}$ | $8.8493 \times 10^{-17}$ | 7.7953 | 1.171 |
| $DM_1(\gamma_0 = \lambda_0 = 0.1)$ | $8.2698 \times 10^{-2}$ | $3.2902 \times 10^{-2}$ | $1.0096 \times 10^{-2}$ | 1.8843 | 1.468 |
| $DM_2(\gamma_0 = \lambda_0 = 0.1)$ | $4.4070 \times 10^{-2}$ | $2.4810 \times 10^{-2}$ | $4.9141 \times 10^{-3}$ | 1.0704 | 1.938 |
| $f_7(x)$ | | | | | |
| $PMM(\gamma_0 = -0.1, \lambda_0 = 0.1)$ | $1.0994 \times 10^{-3}$ | $5.2189 \times 10^{-26}$ | $3.0463 \times 10^{-31}$ | 6.9718 | 1.109 |
| $AM_2(\gamma_0 = \lambda_0 = 0.1)$ | $8.5295 \times 10^{-4}$ | $5.7122 \times 10^{-29}$ | $3.0463 \times 10^{-31}$ | 6.8573 | 1.219 |
| $DM_1(\gamma_0 = \lambda_0 = 0.1)$ | $2.4626 \times 10^{-3}$ | $1.8209 \times 10^{-23}$ | $3.0463 \times 10^{-31}$ | 6.9345 | 0.984 |
| $DM_2(\gamma_0 = \lambda_0 = 0.1)$ | $1.5623 \times 10^{-3}$ | $3.6273 \times 10^{-25}$ | $3.0463 \times 10^{-31}$ | 6.9245 | 1.078 |

## 5. Basins of Attraction

The basins of attraction of the root $t^*$ of $u(t) = 0$ is the set of all initial points $t_0$ in the complex plane that converge to $t^*$ on the application of the given iterative scheme. Our objective is to make use of the basins of attraction to examine the comparison of several root-finding iterative methods in the complex plane in terms of convergence and stability.

On this front, we take a $512 \times 512$ grid of the rectangle $S = [-2, 2] \times [-2, 2] \subset \mathbb{C}$. A colour is assigned to each point $t_0 \in S$ on the basis of the convergence of the corresponding method starting from $t_0$ to the simple root and if the method diverges, a black colour is assigned to that point. Thus, distinct colours are assigned to the distinct roots of the corresponding problem. It was decided that an initial point $t_0$ converges to a root $t^*$ when $\mid t^* - t_0 \mid < 10^{-4}$. Then, point $t_0$ is said to belong to the basins of attraction of $t^*$. Likewise, the method beginning from the initial point $t_0$ is said to diverge if no root is located in

a maximum of 25 iterations. We have used MATLAB R2022a software [28] to draw the presented basins of attraction.

Furthermore, Table 6 lists the average number of iterations denoted by Avg_Iter and the percentage of non-converging points denoted by $P_{NC}$ of the methods to generate the basins of attraction.

**Table 6.** Comparison of the different methods without and with memory in terms of Avg_Iter and $P_{NC}$.

| Without Memory Methods | Avg_Iter | $P_{NC}$ | With Memory Methods | Avg_Iter | $P_{NC}$ |
|---|---|---|---|---|---|
| $p_1(z)$ | | | | | |
| $PM(\gamma = -0.1, \lambda = 0.1)$ | 3.0552 | 0.6718 | $PMM(\gamma_0 = -0.1, \lambda_0 = 0.1)$ | 2.6643 | 0 |
| $SM(\alpha = 10, \gamma = -0.01)$ | 4.1128 | 3.0064 | $AM_2(\gamma_0 = \lambda_0 = 0.1)$ | 2.5278 | 0.0160 |
| $AM_1$ | 3.3635 | 0.0072 | $DM_1(\gamma_0 = \lambda_0 = 0.1)$ | 4.3746 | 7.5332 |
| $CM$ | 3.8199 | 0.2117 | $DM_2(\gamma_0 = \lambda_0 = 0.1)$ | 2.8281 | 0.0084 |
| $p_2(z)$ | | | | | |
| $PM(\gamma = -0.1, \lambda = 0.1)$ | 5.8428 | 10.6179 | $PMM(\gamma_0 = -0.1, \lambda_0 = 0.1)$ | 4.8963 | 4.5556 |
| $SM(\alpha = 10, \gamma = -0.01)$ | 9.4207 | 26.5533 | $AM_2(\gamma_0 = \lambda_0 = 0.1)$ | 4.2219 | 1.9265 |
| $AM_1$ | 9.8161 | 11.2513 | $DM_1(\gamma_0 = \lambda_0 = 0.1)$ | 10.5985 | 33.8319 |
| $CM$ | 6.3409 | 4.5195 | $DM_2(\gamma_0 = \lambda_0 = 0.1)$ | 5.1956 | 0.7041 |
| $p_3(z)$ | | | | | |
| $PM(\gamma = -0.1, \lambda = 0.1)$ | 8.4306 | 21.6465 | $PMM(\gamma_0 = -0.1, \lambda_0 = 0.1)$ | 5.9777 | 3.6148 |
| $SM(\alpha = 10, \gamma = -0.01)$ | 12.8203 | 42.1045 | $AM_2(\gamma_0 = \lambda_0 = 0.1)$ | 6.3765 | 2.2373 |
| $AM_1$ | 10.2165 | 6.6311 | $DM_2(\gamma_0 = \lambda_0 = 0.1)$ | 17.1381 | 63.0899 |
| $CM$ | 9.5562 | 16.9537 | $DM_2(\gamma_0 = \lambda_0 = 0.1)$ | 7.8478 | 3.5973 |

To carry out the desired comparisons, we considered the test problems given below:

**Problem 1.** *The first function considered is $p_1(z) = z^2 - 1$. The roots of this function are 1 and $-1$. The basins corresponding to our proposed method and the existing methods are shown in Figures 1 and 2. From Table 6, it can be seen that the proposed methods, PM and PMM converge to the root in fewer iterations. Furthermore, from the figures, it is observed that PMM converges to the root with no diverging points but the existing methods have some points painted as black. SM, in particular has very small basins.*

**Problem 2.** *The second function taken is $p_2(z) = z^3 - 1$ with roots $-1$, $0.5 + 0.866i$ and $0.5 - 0.866i$. Figures 3 and 4 show the basins for $p_2(z)$ in which it can be seen that SM, $AM_1$ and $DM_1$ have wider regions of divergence. Moreover, the average number of iterations taken by the proposed methods is less in each case compared to the existing methods.*

**Problem 3.** *The third function considered is $p_3(z) = z^4 - 1$ with roots $\pm 1$ and $\pm i$. Figures 5 and 6 show that SM, CM and $DM_1$ have smaller basins. Although PM and PMM have some diverging points, they converge in a fewer number of iterations faster than the existing methods.*

Therefore, we can conclude that from Figures 1–6, it can be observed that $PM$ has larger basins in comparison to $SM$ and $AM_1$ in all cases. The basins for $DM_1$ are very small in comparison to $PMM$ in all cases. In addition, from Table 6, we observe that the average number of iterations taken by the methods $SM$, $AM_1$, and $CM$ are more than $PM$ and for $DM_1$ and $DM_2$, the iterations required are more than $PMM$.

**Remark 3.** *One can see from Figures 1–6 and Table 6 that our proposed methods have larger basins of attraction in comparison to the existing ones. In addition, there is a marginal increase in the average number of iterations per point of the existing methods. Consequently, through our proposed methods, the chances of non-convergence to the root are less when compared to the existing methods.*

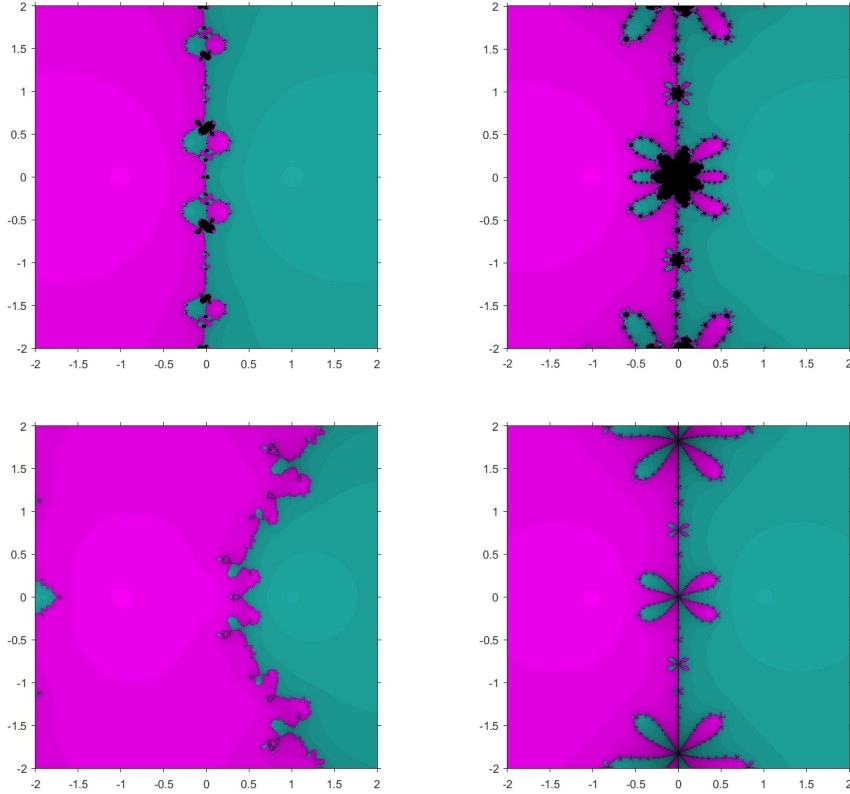

**Figure 1.** Basins of attraction for *PM*, *SM*, *AM*$_1$, and *CM*, respectively, for $p_1(z)$.

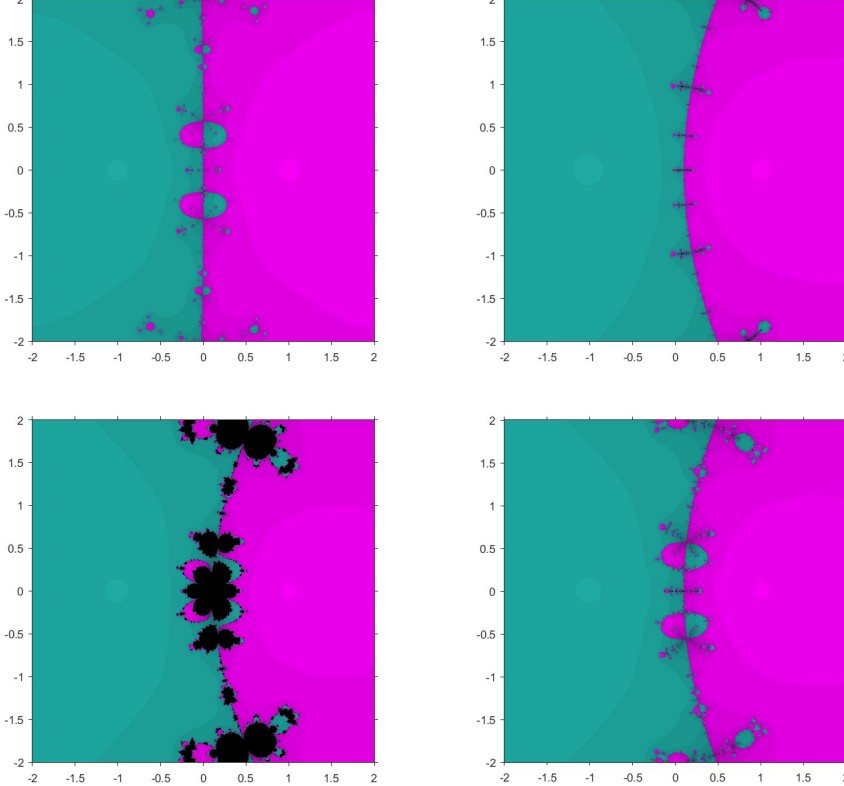

**Figure 2.** Basins of attraction for *PMM*, *AM*$_2$, *DM*$_1$, and *DM*$_2$, respectively, for $p_1(z)$.

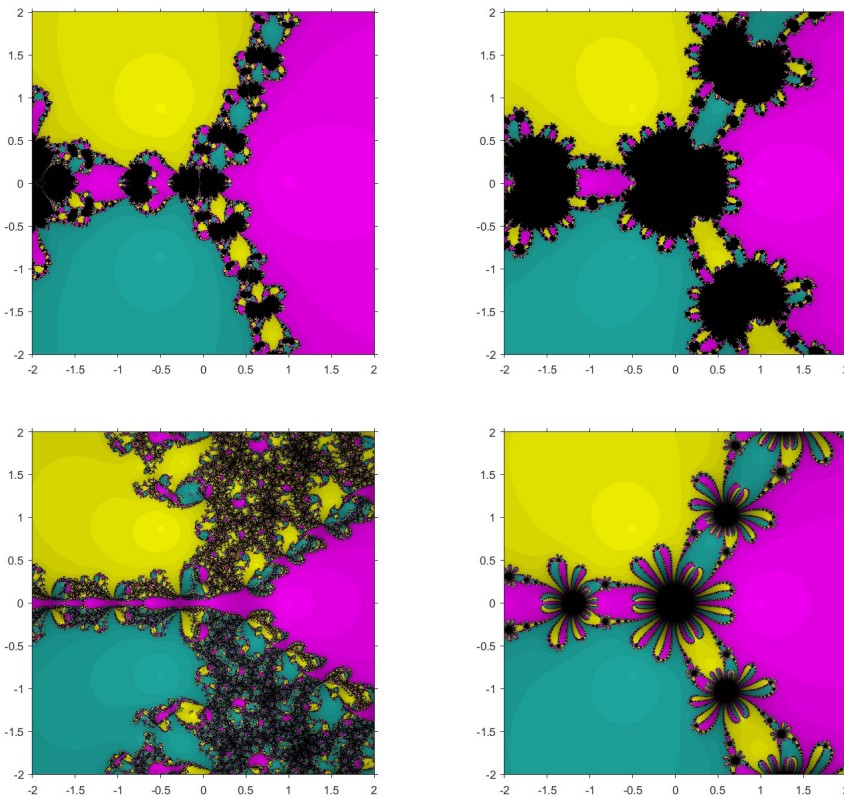

**Figure 3.** Basins of attraction for *PM*, *SM*, $AM_1$, and *CM*, respectively, for $p_2(z)$.

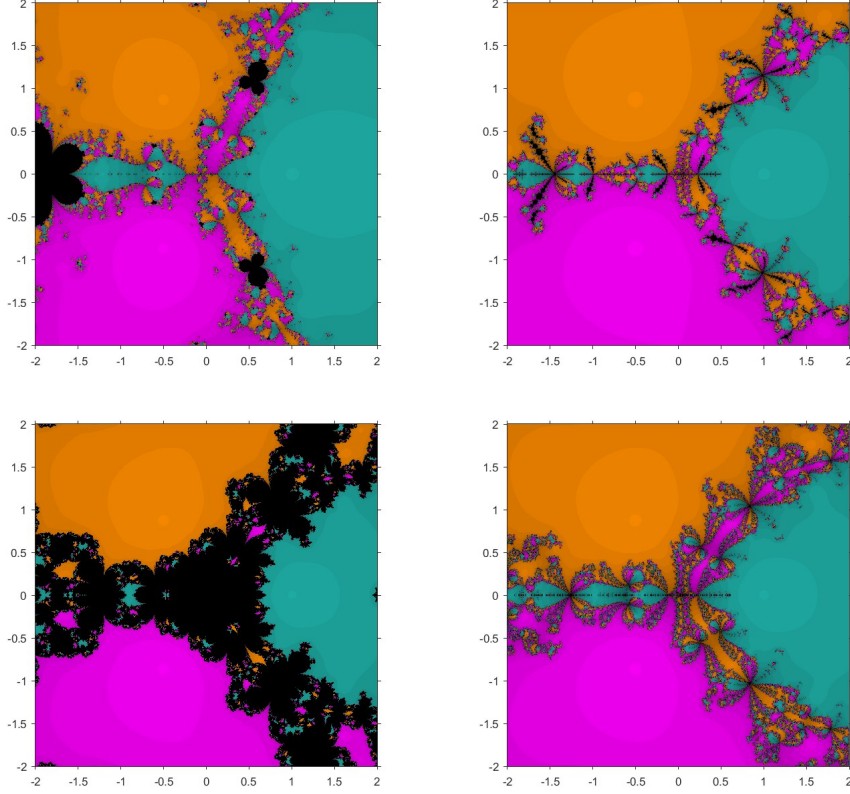

**Figure 4.** Basins of attraction for *PMM*, $AM_2$, $DM_1$, and $DM_2$, respectively, for $p_2(z)$.

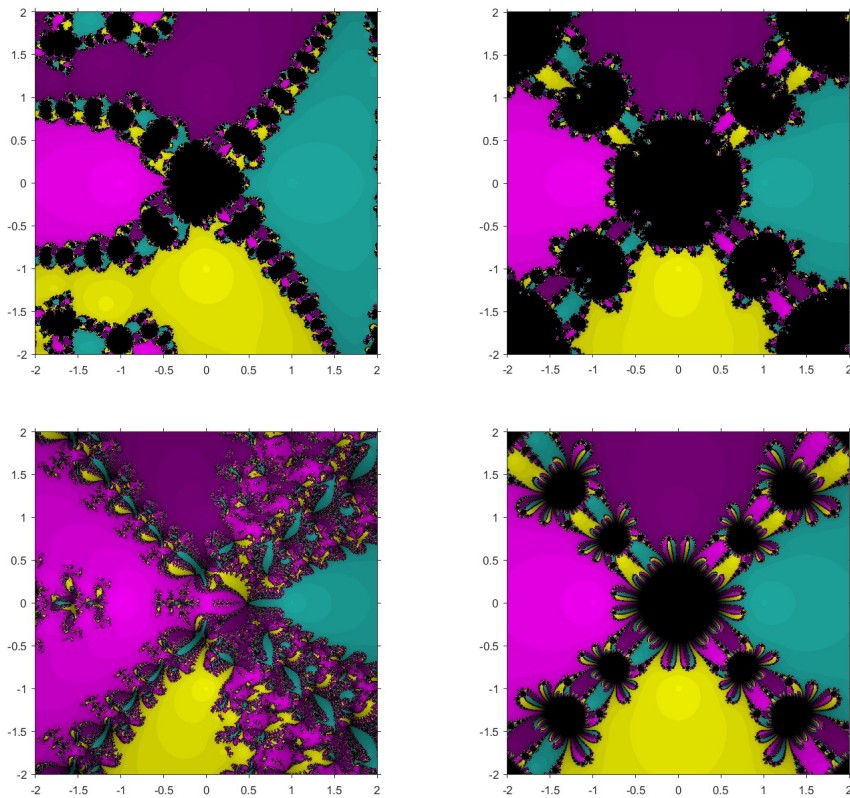

**Figure 5.** Basins of attraction for *PM*, *SM*, *AM*$_1$, and *CM*, respectively, for $p_3(z)$.

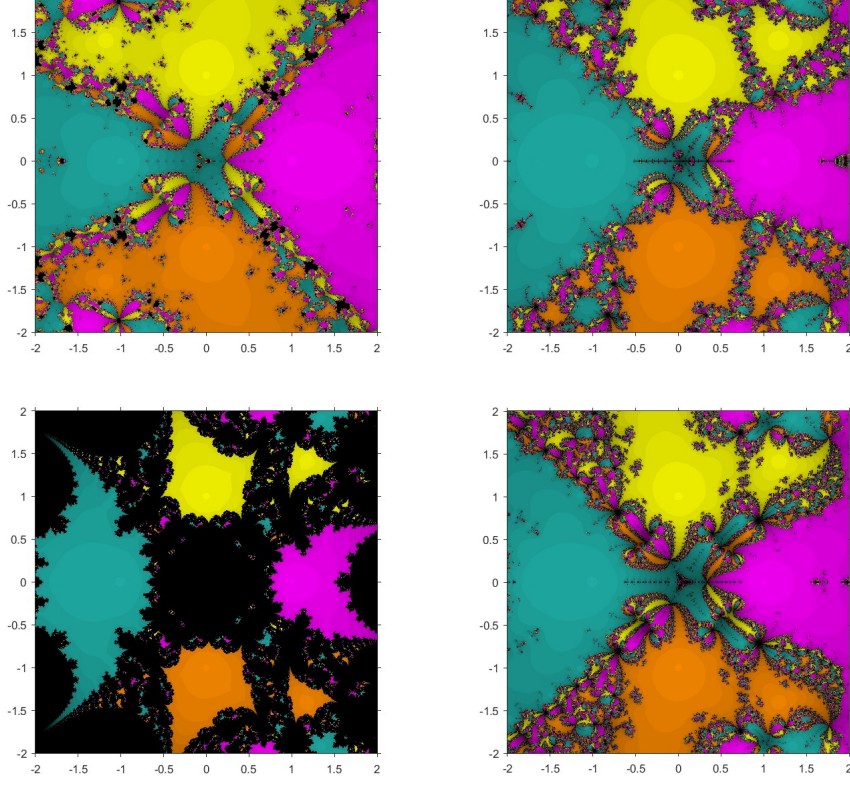

**Figure 6.** Basins of attraction for *PMM*, *AM*$_2$, *DM*$_1$, and *DM*$_2$, respectively, for $p_3(z)$.

## 6. Conclusions

We have proposed a new fourth-order optimal method without memory. In order to increase the order of convergence, we have extended the proposed method without memory to with memory, without the addition of any new functional evaluations taking into consideration two self-accelerating parameters. Consequently, the order of convergence increased from four to seven. Computational results demonstrate that the proposed methods converge to the root with a higher rate in comparison to other methods of the same order at the considered point. In addition, our proposed schemes give results in many of the cases where the existing methods fail in terms of COC and errors. Moreover, we have also presented the basins of attraction for the proposed method as well as some existing methods, which assert that the chances of non-convergence to the root much less in our proposed methods when compared to the existing methods.

**Author Contributions:** M.K.: Conceptualization; methodology; validation; H.S.: writing—original draft preparation; M.K. and R.B.: writing—review and editing, supervision. All authors have read and agreed to the published version of the manuscript.

**Funding:** This project was funded by the Deanship of Scientific Research (DSR) at King Abdulaziz University, Jeddah, Saudi Arabia, under grant no. (KEP-MSc-58-130-43). The authors, therefore, acknowledge with thanks DSR for technical and financial support.

**Institutional Review Board Statement:** Not applicable.

**Informed Consent Statement:** Not applicable.

**Data Availability Statement:** Not applicable.

**Acknowledgments:** The authors would like to sincerely thank the reviewers for their valuable suggestions, which significantly improved the readability of the paper. The second author gratefully acknowledges technical support from the Seed Money Project (TU/DORSP/57/7290) to support this research work of TIET, Punjab.

**Conflicts of Interest:** The authors declare no conflict of interest.

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
