# Peer review of "An Efficient Optimal Derivative-Free Fourth-Order Method and Its Memory Variant for Non-Linear Models and Their Dynamics"

_mca, doi:10.3390/mca28020048_

Round 1

Reviewer 1 Report (Previous Reviewer 1)

The authors proposed a new optimal iterative scheme without memory free from derivatives for solving nonlinear equations. They also provided the basins of attraction to describe a clear picture of the convergence of the proposed method.

Comments:

1.      Give the motivation of the study in detail

2.      What are the limitations of the method?

3.      If possible, give the computational time for convergence of the results in Tables

4.      The literature review is inadequate. The authors may include the following article to update.

(i). https://doi.org/10.1016/j.icheatmasstransfer.2013.04.017

(iI) https://doi.org/10.2298/TSCI191003182Q

(iii). https://doi.org/10.2298/TSCI190504270A

(iv) https://doi.org/10.18869/acadpub.jafm.68.224.24151

(v). https://doi.org/10.1108/HFF-02-2016-0044

(vi). https://doi.org/10.1155/2016/4017076

(vii). DOI:10.24200/sci.2018.20997

Author Response

Reviewer 2 Report (Previous Reviewer 2)

Unfortunately, the authors approached the reviewer's remarks in a very formal way.

These changes are minimal, especially with regard to the scientific novelty of the methods and comparisons with existing ones.

Author Response

Reviewer 3 Report (New Reviewer)

Paper: An Efficient Optimal Derivative-free Fourth Order Method and its Memory Variant for Nonlinear Models and Their Dynamics

In this paper authors propose a new optimal iterative scheme without memory free from derivatives for solving nonlinear equations, increasing order of convergence and giving some numerical examples.

Section 1: Introduction

Have enough information to support the work, and good references, which could be reduced in  number, since some of them, regarding Newton’s Method and its variations, are, as we can say, using a known analogy, “topological transitive” in contents. The structure of this section is ok, regardless theorem’s 1 presence is a little bit strange, but acceptable since the authors states that it will be an important tool. Perhaps a small section after introduction should work better, since gives room to explain the tool and its usefulness.

Section 2: Iterative Method without Memory and its Convergence Analysis

The structure is acceptable, but there’s no need for subsection 2.1. title. This line should be removed. The theorem 2 is the natural continuation of the last paragraph.

On the proof, please rewrite the long math expressions to be split in operation signs and not in variables or numbers. It would be better to understand the calculations made and becomes less confusing.

Section 3: Iterative Method with Memory and its Convergence Analysis

Please remove the line of subsection 3.1. title. There’s no need for it.

Section 4: Numerical Results

The formula for calculating the computational order of convergence have some brackets misplaced.

Section 5: Basins of Attraction

This section is not completely clear how it helps to clarify the strength of the new proposed method and its future contribution to the field. For sure, authors could do better, using the numeric results obtained in section 4 and the computational drawings shown in section 5, discussing the strong points and the weak points of the new method compared with the others.

Section 6. Conclusions

Very weak section, and “Numerical results (…)”presented,  does not “(…) demonstrate that the proposed optimal method and its extension to memory are more effective”. Just point that the authors are in the main road to achieve their goal, but lacks some more organization, work and discussion of the numeric results presented in section 4 and section 5.

Author Response

This manuscript is a resubmission of an earlier submission. The following is a list of the peer review reports and author responses from that submission.

Round 1

Reviewer 1 Report

The authors proposed a new optimal iterative scheme without memory free from derivatives for solving nonlinear equations.

There is nothing new in this method.

The authors said, “There are many iterative schemes existing in the literature which either diverge or fail to work” this is completely wrong.

The Test functions taken in Table 1 are very simple one. It is not a research problem. It is just a graduate exercise.

Reviewer 2 Report

This study is devoted to the actual direction - the search for new approaches for solving nonlinear equations.

The work of the authors is of interest, but there are a couple of questions. First of all, this concerns the clarity of the presentation of the publication:

1. A request to the authors to pay a little more attention to the results and issues of their specific application, to clearly identify areas where they will have advantages in the practical field.

2. It is clear to justify why these methods were chosen for comparison with your approach from many similar ones. Now in the article it is not quite clearly explained.

3. The article contains a fairly large amount of data in tables, figures, but minimal (or even absent) explanations are given, which may reduce the interest of readers who are not as deeply immersed in the topic of the article as the authors.

Reviewer 3 Report

Although the manuscript provides interesting results, the topic is outside the scope of the special issue.   

The manuscript proposes a new optimal iterative scheme without memory free from derivatives for solving nonlinear equations. Local convergence is also proved.

In my opinion, it is interesting and can be recommended for publication after the following comments are addressed. 

  1. A number of high-order iterative methods have been developed in past two decades. But only a few methods are mentioned in the current manuscript. More relevant works should be commented and compared.
  2. Semi-local convergence is an important issue. I would like to see such results for their methods.

Round 2

Reviewer 1 Report

The motivation of the present study is not clear.

The Test functions taken in Table 1 are very simple functions. It is not a research problem. It is just a graduate exercise.

1.      There are several papers published in iterative methods in literature, What is new in this paper? Give in detail.

The literature survey  is inadequate.

Reviewer 3 Report

I have no further comments.